# Changes in human peripheral blood mononuclear cell (HPBMC) populations and T-cell subsets associated with arsenic and polycyclic aromatic hydrocarbon exposures in a Bangladesh cohort

Fredine T. Lauer[1], Faruque Parvez[2], Pam Factor-Litvak[3], Xinhua Liu[4], Regina M. Santella[2], Tariqul Islam[5], Mahbubul Eunus[5], Nur Alam[5], A. K. M. Rabiul Hasan[5], Mizanour Rahman[5], Habibul Ahsan[6], Joseph Graziano[2], Scott W. Burchiel[1]*

1 Department of Pharmaceutical Sciences, University of New Mexico College of Pharmacy, Albuquerque, NM, United States of America, 2 Department of Environmental Health Sciences, Mailman School of Public Health, Columbia University, New York, NY, United States of America, 3 Department of Epidemiology, Mailman School of Public Health, Columbia University, New York, NY, United States of America, 4 Department of Biostatistics, Mailman School of Public Health, Columbia University, New York, NY, United States of America, 5 University of Chicago Field Research Office, Dhaka, Bangladesh, 6 Department of Health Studies, University of Chicago, Chicago, IL, United States of America

* sburchiel@salud.unm.edu

**Data Availability Statement:** The data set for this study can be accessed via the following DOI: 10.

## Abstract

Exposures to environmental arsenic (As) and polycyclic aromatic hydrocarbons (PAH) have been shown to independently cause dysregulation of immune function. Little data exists on the associations between combined exposures to As and PAH with immunotoxicity in humans. In this work we examined associations between As and PAH exposures with lymphoid cell populations in human peripheral blood mononuclear cells (PBMC), as well as alterations in differentiation and activation of B and T cells. Two hundred men, participating in the Health Effects of Arsenic Longitudinal Study (HEALS) in Bangladesh, were selected for the present study based on their exposure to As from drinking water and their cigarette smoking status. Blood and urine samples were collected from study participants. We utilized multiparameter flow cytometry in PBMC to identify immune cells (B, T, monocytes, NK) as well as the T-helper (Th) cell subsets (Th1, Th2, Th17, and Tregs) following *ex vivo* activation. We did not find evidence of interactions between As and PAH exposures. However, individual exposures (As or PAH) were associated with changes to immune cell populations, including Th cell subsets. Arsenic exposure was associated with an increase in the percentage of Th cells, and dose dependent changes in monocytes, NKT cells and a monocyte subset. Within the Th cell subset we found that Arsenic exposure was also associated with a significant increase in the percentage of circulating proinflammatory Th17 cells. PAH exposure was associated with changes in T cells, monocytes and T memory (Tmem) cells and with changes in Th, Th1, Th2 and Th17 subsets all of which were non-monotonic (dose dependent). Alterations of immune cell populations caused by environmental exposures to

6084/m9.figshare.8144492 (doi.org/10.6084/m9.figshare.8144492).

**Funding:** This work was supported by the National Institutes of Environmental Health Sciences (NIEHS) ViCTER Program R01 ES019968S1 (SWB) and by the Columbia NIEHS Superfund Basic Research Program P42 ES010349 (JG) and P30 ES009089 (JG).

**Competing interests:** The authors have declared that no competing interests exist.

As and PAH may result in adverse health outcomes, such as changes in systemic inflammation, immune suppression, or autoimmunity.

## Introduction

Arsenic exposure is prevalent worldwide and occurs primarily through consumption of naturally contaminated ground water and to a lesser degree through food and air. Inorganic arsenite (trivalent, +3) and arsenate (pentavalent, +5) are found in ground water in areas with abundant surrounding natural sources. The Health Effects Arsenic Longitudinal Study (HEALS) cohort, in Araihazar, Bangladesh was established to evaluate the effects of inorganic As exposure on various health outcomes. This cohort of over 35,000 men and women live in rural regions with highly variable concentrations of inorganic As in household well water and are at increased risk of various cancers, diabetes, and cardiovascular and respiratory disease. In particular, the rates of skin, kidney and bladder cancer are increased [1–3]. Increased cardiovascular and pulmonary morbidity has also been found in Bangladesh associated with As exposure [4–10].

PAHs are produced during the burning of fossil fuels and other organic matter, and are found in tobacco smoke. Humans are exposed to PAHs (volatile, semi-volatile, and non-volatile species), some of which adsorb to airborne particulate matter (PM) [11]. In an earlier study in Bangladesh, people exposed to urban traffic pollution were found to have high PAH exposures [12]. Cigarette smoke contains numerous PAHs and is a well-established source of exposure. In humans, PAHs have been associated with cancer [13], suppression of the immune system [14, 15], and lung and airway disease [16, 17]. PM exposures have been associated with cardiovascular disease and mortality [18]. In Bangladesh it is quite common for people to experience combined exposure to As and PAHs through everyday activities.

In our previous work in Bangladesh, we found disparate effects of As and PAH exposures on immune parameters in a cohort of 197 men. Arsenic was positively associated with proinflammatory cytokine production, most notably IL-1β [19]. PAH exposure was associated with suppression of T cell proliferation (TCP) and the inhibition of secretion of several cytokines, including IFNγ, IL-2, IL-10, and IL-17A. We did not detect an interaction between urinary As and PAH exposure (measured by PAH-DNA adducts) for cytokine production. While As and PAHs exert both genotoxic and non-genotoxic effects, the mode of action of these environmental agents, at least for immune function, appears to be quite different.

Our work in mice has shown that the non-genotoxic effects of As and PAHs are largely mediated through alterations in cell activation signaling pathways [20–22]. For genotoxicity, As has been shown to inhibit DNA repair via binding to Zinc finger proteins, such as poly ADP-ribose polymerase (PARP) [23–26]. Since large PAHs, such as benzo[a]pyrene (BaP) are complete carcinogens and known to induce DNA damage, we postulated that they might act synergistically with As in humans. Indeed, in *in vivo* animal models at some doses, there is a synergy between As and PAHs [27], and this synergy is easily observed in the thymus following *in vitro* exposures [28]. However, following chronic exposure to As in men, we found no evidence of synergy with PAHs for TCP or cytokine production in PBMC [19]. Thus, in this same cohort of men, we further characterize the immune effects of chronic exposures to As and PAHs in PBMC.

Using multiparameter (11 color) flow cytometry, we examined the associations of As and PAH exposures, alone or in combination, with cell surface markers (CSM) on PBMC and Th

functional subsets. We assessed CSM for B and T lymphocytes, monocytes, Tmem and NK cells, as well as B cell activation. Through intracellular cytokine staining (ICS) of viable cells and multiparameter flow cytometry, we also assessed the effects of As and/or PAHs on the differentiation of functional subsets of Th cells, including Th1, Th2, Th17, and T regulatory (Tregs) cells following anti-CD3/anti-CD28 activation.

## Methods

### Recruitment and consent of study participants

The Institutional Review Board (IRB) of Columbia University approved the study protocol and the Bangladesh Medical Research Council (BMRC) gave ethical clearance. Recruitment materials and consent forms were translated into Bengali and then back translated into English. For men unable to read, a village health worker from the area read the informed consent and in simple terms explained procedures in the presence of a witness. The study participants provided either written or verbal consent. University of New Mexico's (UNM) IRB approved a protocol for the analysis of the biological samples.

For participation in this study healthy adult males from the HEALS cohort were identified. We chose to recruit only males for this study since smoking rates in females in Bangladesh is extremely low, approximately 2.3% compared to males, approximately 44% [29]. Recruitment was based on well water As concentrations and smoking status in men age 35–75 years. Since the goal of this study was to examine the combined effects of As and PAH, half of the participants were identified as drinking from wells with water concentrations of As > 50 ppb with the remainder < 50 ppb. Cigarette smoking results in exposure to PAHs; approximately half the individuals in each water As strata were current smokers. Men with illnesses related to immune dysfunction and/or taking medications that might influence immune function, including those for cardiovascular disease and diabetes were not included.

Three hundred seventeen eligible participants were identified through the HEALS database, of which 246 visited the study clinic. Blood and urine samples were successfully obtained and PBMC isolated from 200. Some samples did not have an adequate volume for PAH-DNA analysis and some had an inadequate number of cells for cell surface marker or intracellular staining assays. Thus, 179 samples were assayed for cell surface markers and 180 samples for intracellular cytokines.

### Quantification of urinary arsenic and creatinine

As described in Parvez et al., (2019) spot urine samples were collected in 50 mL acid-washed tubes then stored at −80°C prior to analyses. Total urinary arsenic (UAs) was quantified by graphite furnace atomic-absorption spectrophotometry (GFAAS) using a Perkin-Elmer Analyst 600 graphite furnace system as previously described [30]. The detection limit for UAs was 2 µg/L. Urinary creatinine was quantified by a colorimetric method based on the Jaffe reaction. Total urinary As adjusted for creatinine was reported as microgram UAs per gram Cr (µg UAs/g Cr).

### Quantification of polycyclic aromatic hydrocarbon DNA (PAH-DNA) adducts

PAH-DNA adducts in DNA isolated from HPBMC were analyzed by competitive ELISA, using methods described previously [31, 32]. Briefly, 96 microwell plates were coated with 2 ng of benzo[a]pyrene diol epoxide (BPDE)-I-DNA (5 adducts/$10^3$ nucleotides). A previously characterized rabbit antiserum was used at 1:500,000 dilution to detect adducts [33]. A

standard curve consisted of diluted rabbit antiserum with BPDE-I-DNA in carrier nonmodi-fied calf thymus DNA such that 50 µl contained 0.08–200 fmol BPDE-I-deoxyguanosine adduct in 50 µg DNA. Following sonication, denaturation and cooling on ice samples were assayed at 10 µg/well. Biotinylated goat anti-rabbit IgG-alkaline phosphatase (Boehringer Mannheim, Indianapolis, IN) was used at 1:5000 and avidin-alkaline phosphatase at 1:6666. CDP-star with Emerald (Tropix Bedford, MA) was used and chemiluminescence measured with a TR717 Microplate Luminometer (PE Applied Biosystems, Foster City, CA) at 542 nm. Samples displaying <15% inhibition were considered non-detectable and assigned a value of 1 adduct/$10^8$ nucleotides, an amount midway between the lowest positive value and zero. For quality control, a 5% blinded duplication analysis was performed on those samples from sub-jects with the most DNA available.

## Collection and cryopreservation of peripheral blood mononuclear cells (PBMC)

For collection and cryopreservation of PBMC the laboratory personnel in Bangladesh followed the detailed procedure previously published by Lauer et al. [34]. The samples used for assays in this study were the same samples described in Parvez et al. [19]. Briefly, blood samples were diluted with phosphate-buffered saline (DPBS⁻; Sigma-Aldrich, D8537) then layered over Fico/Lite-LymphoH (Atlanta Biologicals, I40150) and centrifuged. The mononuclear cell layer was collected and washed twice with cold DPBS⁻. Cell pellets were resuspended and cryopre-served in Freezing Media (Athena Enzyme Systems, 0406).

Cryopreserved HPBMC samples were transported to the United States using liquid nitro-gen-charged Dry Shippers from Cryoport, Inc. (Irvine, CA). The internal temperature of the Dry Shipper remained < -180˚C and was monitored remotely. Once the shipper and samples arrived at UNM they were transferred to liquid nitrogen where they remained prior to thawing for assays.

## Thawing of HPBMC

Samples were thawed based on the procedures previously described by Lauer et al. [34]. Assays consisted of batches of approximately 20 samples. Samples were thawed quickly and washed three times in complete RPMI [RPMI-HEPES modified (Sigma- Aldrich, R5886), with 10% FBS, 1% of 200 mM L-glutamine (Gibco, 25030–081) and 1% of 10,000 U/ml penicillin and 10,000 µg/ml streptomycin (Pen/Strep; Gibco, 15140–122)], referred to here on as cRPMI, then resuspended for counting. Cells were counted on the Nexcelom Cellometer Auto 2000 Cell Viability Counter using acridine orange and propidium iodide (AO/PI; Nexcelom Biosci-ence, CS2-0106) in accordance with manufacturer's directions. Samples with viability greater than 80% were used for analysis. Cells were aliquoted for either cell surface marker (CSM) staining or plated in a sterile petri dish for intracellular marker (ICM) staining. ICM samples were placed in a 37˚C, humidified, 5% $CO_2$ incubator overnight to "rest".

## Cell surface marker staining and detection

Antibodies and fixable viability stain (FVS) were acquired from BD Biosciences (S1 Table). One million cells were aliquoted into each of two cluster tubes (Fisher 07-200-31), one for stained sample and the other for unstained sample, containing 200 µl staining buffer [Dulbec-co's Phosphate Buffer without calcium or magnesium, 0.2% heat inactivated FBS and 0.09% sodium azide] then centrifuged 10 min at 250xg at 4µC. Samples were washed once more and resuspended in 50 µl of BD's Horizon Brilliant Stain Buffer (Cat. No. 563794). A cocktail of CD16, HLA-DR, CD56, CD19, CD8, CD127 CD4, FVS 780, CD45RO, CD3 and CD14, in

amounts indicated by the manufacturer (S1 Table), was prepared and aliquoted into each sample designated to be stained. Staining buffer was added to the unstained samples. Samples were incubated on ice, in the dark for 20 min. Following incubation, cells were washed with 500 μl cold stain buffer, centrifuged, fixed with Cytofix (Cat. No. 554655), then incubated on ice in the dark for 30 min. Following fixation, cells were washed with 500 μl stain buffer then resuspended in 500 μl stain buffer and held at 4°C in the dark until analyzed on the LSR Fortessa (BD Biosciences; San Jose, CA), flow cytometer using the blue, yellow/green, red and violet lasers and BD FACSDiva software v6. The gating strategy employed for identifying immune cell subsets was: T-cells (CD3+CD19-); Th cells (CD3+CD4+CD8-); cytotoxic T lymphocytes (CTL; CD3+CD4-CD8+); T-memory cells (CD3+CD45RO+); monocytes (classically defined as CD14+CD16-); monocytes (defined as CD14+CD16+); B-cells CD3-CD19+; activated B-cells CD19+HLA-DR+; natural killer cells (NK; CD3-CD56+); NKT cells (CD3+CD56+) and cells expressing the IL-7 receptor alpha (IL-7Rα; CD127+) (S1 Fig [34]).

Prior to sample analysis of CSM or ICM, we ran CS&T beads (Cat no. 641319) described by Lauer et al. [34]. Color compensation samples were ran prior to each batch analysis and compensation was established by FACSDiva software v6. Experiment FCS files from FACSDiva were imported into FlowJo v10 software for gating analysis. Fluorescence minus one (FMO) approach was used to establish gates for the markers prior to the start of the study.

## Intracellular marker staining and detection

Detailed procedures for ICM staining described by Lauer et al. [35] were followed to identify T cell and Th lymphocytes and the intracellular targets, IFNγ, IL-4, IL-17A and Foxp3, and to indicate the Th cell subsets Th1, Th2, Th17 and Treg. Following a resting period of approximately 24 hr, the cells were collected, centrifuged and resuspended at $5.6x10^6$ cell/ml. Cells were then plated in flat bottom 96 well plates that had been coated with anti-CD3 (clone OKT3; eBiosciences Cat. No. 16-0037-85) antibody or DPBS⁻ (no stimulation).

One set of stimulated wells (CD3 coated) was to be stained with the cocktail of antibodies, one set of stimulated wells (CD3 coated) was to be unstained (DPBS⁻ + FVS) and one set of unstimulated (DPBS⁻) wells was to be stained with the antibody cocktail. The co-stimulant anti-CD28 was added to each of the CD3 coated wells to yield a final concentration of 2 μg/well. Plates were placed in a humidified, 37°C, 5% $CO_2$ incubator overnight (18–24 hr). Following incubation, Brefeldin A Ready Made Solution (BFA; 10 mg/ml; Sigma-Aldrich, Cat. No. B5936) was added to each well for a final concentration of 10 μg/ml and the plate was returned to the incubator for an additional 4 hr.

Following incubation samples were transferred from 96 well plates to cluster tubes, washed once with staining buffer, centrifuged, and resuspended following the removal of staining buffer. Brilliant stain buffer was added to each tube as was the CSM antibody cocktail (CD127, CD4, CD69, CD45RO, CD3, CD25 and FVS620), prepared following the manufacturer's recommended amounts (S2 Table), to the 'stained' samples. To the 'unstained' samples staining buffer plus viability stain was added. Samples were incubated 20 min on ice in the dark. Following the staining, samples were washed once with stain buffer then fixed and permeabilized using BD Pharmingen Transcription Factor Buffer set (TF; BD Biosciences Cat. No. 562574) according to manufacturer's directions. Samples were incubated on ice for 45 min in the dark. Following fixation and permeabilization, samples were washed twice with TF perm/wash buffer (included in the TF Buffer set) according to manufacturer's directions. Samples were resuspended in Brilliant Stain Buffer and a cocktail ICM was added to the 'stained' samples, stain buffer was added to the 'unstained' samples. The ICM cocktail was prepared following the manufacturer's recommended amounts (S2 Table) and included IFNγ, IL-17A, IL-4, and

Foxp3. Samples were incubated for 45 min on ice in the dark. All samples were washed twice with TF perm/wash buffer, resuspended with staining buffer, protected from light and held at 4˚C until run on the LSRFortessa flow cytometer. As described above in, 'Cell surface staining and detection by flow cytometry' section, we utilized color compensation samples and CS&T beads in setting up for analysis of the samples.

We used established antibody panels, gating strategies and analytical approaches, published previously by Lauer et al. [35]. For consistency, data analysis was performed at the end of the study. The preliminary gating strategy (S2 Fig) was to gate on HPBMC (SSC and FSC), single cells (FSC height and FSC area) and then live cells. Live cells were the unstained population as the dead cells were stained with the FVS. The use of a FVS was extremely useful in gating-out or eliminating dead cells which tend to nonspecifically bind cell surfaces and intracellular antibodies. Next we gated T cells, identified by the markers CD3 (T cell) and CD4 (Th1 cell). T cell subsets were identified by intracellular staining and included: T-regulatory (CD3+CD4+Foxp3 +CD25+), Th1 (CD3+CD4+IFNγ+), Th2 (CD3+CD4+IL4+), Th17 (CD3+CD4+IL17A+); additionally CD69 was used to detect cell nonspecific staining; none was detected in our analysis. We have provided antibody catalog information, manufacturer recommended amounts for staining, as well as the laser and filters used for T cell subset discrimination (S2 Table).

## Statistical analysis

Statistical analysis was performed separately for cells surface markers (CSM) and intracellular marker staining (ICM) for T cell subsets. Statistical analysis followed the procedures previously described by Parvez et al. [19]. Briefly, we calculated descriptive statistics for all variables; including CSM (phenotypic markers) and ICM (T cell subsets), PAH-DNA adducts and urinary arsenic concentrations (UAs/Cr; total urinary As adjusted for creatinine). Spearman correlation coefficients were calculated to describe bivariate associations among the continuous variables. We imputed the missing value for body mass index (BMI) for one participant based on a regression model for BMI with predictors of age and smoking status. Age, BMI and smoking status were included as covariates in all statistical models. To reduce the impact of extreme variables and meet model assumptions, we transformed variables with right skewed distributions.

The Generalized Additive Model (GAM) was applied to evaluate possible non-monotonic relationships between exposures (UAs/Cr and PAH-DNA adducts) and each cell surface as the outcome. We used SAS version 9.4 and R version 3.5.1 for statistical analysis. GAM allows both parametric and non-parametric components to determine non-monotonicity as well as non-linear patterns of associations between exposure and outcome variables. Based on the results of the GAM models, we fit linear models for the immune parameters with the most parsimonious polynomials of the exposure variables to describe the patterns of associations. In these models, we used the likelihood ratio test to detect the overall effect of exposure (linear and higher order terms as appropriate). We calculated the change in R-squared ($\Delta R^2$) to demonstrate the overall effect size of the exposure as the interpretation of the estimated individual regression coefficients of polynomials of the exposure variable is difficult. To adjust for multiple testing on CSM (11 phenotypes) or ICM (8 T cell subsets), we used the Benjamini and Hochberg method to control for false discovery rate (FDR). Relevant associations were those having p-values <0.05 and FDR <0.05.

To examine whether UAs/Cr modified the association between PAH-DNA adducts and the CSM and ICM, we stratified the sample by the median of UAs/Cr and fit linear models with covariates and polynomials of PAH-DNA adducts, as suggested by the GAM. We used the Wald test to detect differences in the coefficients of PAH-DNA adducts variables between

UAs/Cr strata above and below the median and then adjusted for multiple tests in each set of immune markers (CSM and ICM). The data set for this study can be accessed at 10.6084/m9. figshare.8144492.

# Results

## Characteristics of the study sample

Characteristics of the study participants are shown in Table 1. The majority of participants were over 50 years old (66%), and 75% had a BMI less than 25. By design, half of the study participants were active smokers (49%) and 49% were exposed to water As >50 μg/L, the nationally accepted limit in Bangladesh, unlike the WHO standard of 10 μg/L. Descriptive data of immune cell phenotypes (reported as percentage of live cells) as well as T cell subset phenotypes (reported as percentage live cells and % of CD4+ cells) are also reported in Table 1. Isolated PBMC include lymphocytes (T, B and NK cells) and monocytes; the frequencies of these populations typically range from 79–90% lymphocytes and 10–20% monocytes [36–38]. Isolated HPBMC were approximately 84% lymphocytes (T cells 59%, B cells 12%, and NK cells

**Table 1. Demographics, exposure, immune cell phenotypes and T cell subsets of a male Bangladeshi Study Cohort.**

| Variables | Mean (SD) | Median (range) |
|---|---|---|
| Demographics (n = 180) | | |
| Age | 51.7 (6.3) | 52 (36, 65) |
| BMI | 22.1 (3.8) | 21.5 (13.7, 34.7) |
| Ever smoked | 49% | - - - |
| Exposure | | |
| PAH-DNA adducts (per $10^8$ nucleotides) | 2.2 (1.4) | 1.8 (0.5, 8.0) |
| UAs/Cr (ug/g Cr) | 162.3 (180.3) | 96.4(10.0, 1116.0) |
| Cell phenotypes (n = 179) % live cells | | |
| T cell (CD3+) | 58.9 (9.5) | 59.9 (13.4, 81.4) |
| Th cell (CD3+ CD4+) | 32.7 (7.4) | 32.7 (13.4, 58.2) |
| CTL (CD3+CD8+) | 21.5 (7.8) | 19.9 (6.9, 49.3) |
| B cell (CD19+) | 12.0 (4.8) | 10.7 (3.0, 29.5) |
| Monocyte (CD14+CD16-) | 8.6 (4.5) | 7.9 (1.1, 20.7) |
| T memory (CD3+CD45RO+) | 33.5 (7.8) | 33.6 (16.1, 55.1) |
| Activated B cell | 11.1 (4.7) | 10.2 (2.9, 28.4) |
| Monocytes (CD14+CD16+) | 1.3 (1.0) | 1.0 (0.1, 6.3) |
| NK (CD3-CD56+) | 12.8 (6.9) | 11.0 (2.3, 34.9) |
| NKT (CD3+CD56+) | 3.6 (3.6) | 2.6 (0.2, 33.9) |
| IL-7 receptor α | 36.8 (7.3) | 36.3 (18.9, 54.0) |
| T cell subsets (n = 180) % of CD4+ cells | | |
| Th cell (CD3+C4+) | 39.4 (8.4) | 39.3 (10.9, 59.2) |
| Th1 cell (CD3+CD4+IFNγ+) | 4.4 (3.1) | 3.7 (0.02, 17.8) |
| Th2 cell (CD3+CD4+IL4+) | 2.3 (2.5) | 1.2 (0.4, 20.3) |
| Treg cell (CD3+CD4+Foxp3+CD25+) | 14.6 (4.6) | 14.0 (3.2, 27.0) |
| Th17 cell (CD3+CD4+IL17A+) | 3.6 (3.6) | 2.7 (0.09, 23.3) |
| Stimulated CD3+ (CD3+ CD69+CD25+) % live cells | 44.2 (10.8) | 45.5 (1.0, 69.7) |
| Stimulated CD4+ (CD3+CD4+ CD69+CD25+) % live cells | 30.3 (8.6) | 29.9 (0.7, 49.1) |
| Stimulated Live cells (CD69+CD25+) % live cells | 52.5 (12.5) | 53.6 (2.1, 75.4) |

13%) and 9% monocytes. We also examined the expression of the IL-7 receptor on cell subsets because we have previously shown that As interferes with IL-7 signaling in mice [21, 22].

We observed negative correlations between T cells and B cells, activated B cells, monocytes, and NK cells which ranged from -0.198 to -0.601. T cells correlated with CTL, cells with IL7Rα, NKT, Th cells and Tmem cells with Spearman correlation coefficient (r) ranged from 0.236 to 0.638. There was no association between UAs/Cr and PAH-DNA adducts (r = 0.007, p = 0.93).

## Associations between exposures and cell surface markers

Results of the regression analyses for associations with As exposure and immune cell phenotypes (CSM) are shown in Table 2. We found associations between UAs/Cr and Th cells (CD3+CD4+), monocytes (CD14+), non-classical monocytes (CD14+CD16+) and NKT cells (CD3+CD56+) in models adjusted for age, BMI, smoking status and PAH-DNA adducts. Th cells (CD4+) increased with As exposure (Fig 1A), whereas the associations for monocytes (both CD14+ and CD14+CD16+) and NKT cells were non-monotonic (Fig 1B–1D). Associations between PAH exposure and cell surface phenotypes are summarized in Table 3. We found associations between T cells (CD3+), Th cells (CD3+CD4+), monocytes (CD14+) and Tmem cells (CD3+CD45RO+) after adjustment for age, BMI, smoking and UAs/Cr. For T, Th, Tmem cells, and monocytes there were non-monotonic associations, as illustrated in Fig 2.

## Associations between exposures and T cell subsets

Associations between UAs/Cr and T cell subsets are shown in Table 4. There was a positive linear association between UAs/Cr and Th17 cells after adjustment for age, BMI, smoking and PAH-DNA adducts as illustrated in Fig 3. Associations between PAH-DNA adducts and T cell subsets and activated subsets are shown in Table 5. There were non-monotonic associations between PAH exposure and Th, Th1, Th2 and Th17 cells after adjusting for age, BMI, smoking status and UAs/Cr (illustrated in Fig 4). In the analysis of UAs/Cr and PAH-DNA adduct

**Table 2. Association between urinary arsenic concentration per creatinine (UAs/Cr) and immune cell phenotypes (n = 179).**

| Cell phenotype | [a]ΔR$^2$ (%) | [b]p-value | [c]FDR | [d]B$_1$ (se) | [d]B$_2$ (se) | [d]B$_3$ (se) |
|---|---|---|---|---|---|---|
| T cell | 3.12 | 0.05 | 0.10 | 1.69 (0.84) | -1.37 (0.65) | --- |
| Th cell | 3.86 | 0.004 | 0.02 | 1.72 (0.60) | --- | --- |
| log(CTL) | 0.07 | 0.73 | 0.73 | -0.01 (0.03) | --- | |
| log(B cell) | 0.31 | 0.41 | 0.58 | 0.03 (0.03) | --- | |
| Monocyte (CD14+) | 7.70 | 0.002 | 0.02 | -1.24 (0.57) | 0.42 (0.32) | 0.61 (0.21) |
| T memory cell | 0.16 | 0.58 | 0.70 | 0.37 (0.66) | --- | --- |
| log(Activated B cell) | 0.06 | 0.71 | 0.73 | -0.01 (0.03) | --- | |
| log(Monocyte CD14+CD16+) | 6.50 | 0.009 | 0.03 | 0.17 (0.10) | 0.16 (0.06) | 0.06 (0.04) |
| log(NK) | 1.73 | 0.077 | 0.14 | -0.09 (0.05) | --- | --- |
| log(NKT) | 4.95 | 0.011 | 0.03 | 0.05 (0.08) | -0.19 (0.06) | --- |
| IL7α Receptor (n = 178) | 0.37 | 0.42 | 0.58 | -0.52 (0.65) | --- | |

Note: All estimates are from linear models with polynomials of exposure variable X, adjusted for age, BMI, ever smoked and effect of PAH-DNA adducts.

[a]ΔR$^2$: The change in R$^2$ for percent of variation in outcome explained by the effect of UAs/Cr adjusting for other variables.

[b]p-value was from likelihood ratio test for the effect of UAs/Cr.

[c]FDR: False discovery rate

[d]B$_1$: Estimated coefficient of X; B$_2$: estimated coefficient of X$^2$; B$_3$: estimated coefficient of X$^3$; for X = log(UAs/Cr /96.2963).

se: Standard error

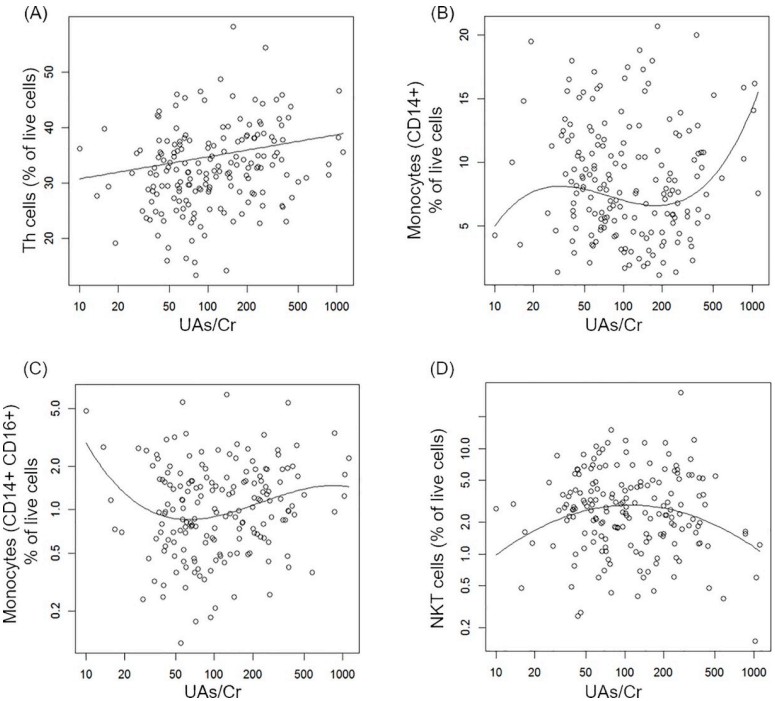

**Fig 1. Covariate-adjusted associations between UAs/Cr and cell surface markers.** Individual data points represent observations; solid lines represent estimated mean outcome vs. UAs/Cr adjusted for other variables in the model. The outcomes (A-B) were Th (CD4+) cells and monocytes (CD14+) and outcomes (C-D) were a subset of monocytes (CD14+CD16+) and NKT cells in logarithmic scales.

interaction, we did not find that the covariate-adjusted associations between PAH-DNA adducts and CSM or T cell subsets differed by the strata of UAs/Cr above and below the median. The results from the models with covariates of age, BMI and smoking status was similar to that of the models further controlling for stratum specific UAs/Cr.

## Discussion

In a recent study, we showed that As and PAH exposures were associated with changes in T cell proliferation and cytokine production in a male cohort in Bangladesh. Arsenic and PAH exposures were associated with dose-related and overall increased secretion of IL-1β. IL-1β is a pro-inflammatory cytokine and is a key mediator of the inflammatory response [39]. Additionally, PAH exposure was associated with suppression of T-cell proliferation and dose-related changes in IFNγ, IL-2, IL-10, and IL-17A cytokine production [19]. In this study we investigated the associations between the As and PAH exposures on HPBMC immune markers assessed *ex vivo* using flow cytometry in the same samples reported in Parvez et al. [19].

For CSM, we found that As exposure was associated with an increase in the percentage of Th cells. Additionally, As demonstrated a trend towards a non-monotonic increase with CD14 + monocytes and CD14+CD16+ cells (non-classical or intermediate monocytes). This supports our previous finding of increased IL-1β secretion as it is produced by cells such as monocytes and activated macrophages [39]. There was a non-monotonic association between UAs/Cr and NKT showing an increase at lower As levels and a decrease at higher amounts. PAH adducts were negatively associated with the presence of T and Tmem cell phenotypes in PBMC. A decrease in T, Th, and Tmem cells would be consistent with our earlier finding of suppressed T cell proliferation for PAH adducts.

**Table 3. Association between PAH-DNA adducts and immune cell phenotypes (n = 179).**

| Cell phenotype | [a]$\Delta R^2$ (%) | [b]p-value | [c]FDR | [d]$B_1$ (se) | [d]$B_2$ (se) | [d]$B_3$ (se) |
|---|---|---|---|---|---|---|
| T-cell | 9.88 | <0.0001 | 0.0004 | -2.19 (1.12) | -5.37 (1.22) | - - - |
| Th-cell | 14.45 | <0.0001 | <0.0001 | -4.49 (1.75) | -4.19 (0.96) | 2.79 (1.36) |
| log(CTL) | 0.10 | 0.68 | 0.68 | 0.02 (0.04) | - - - | - - - |
| log(B cell) | 1.29 | 0.09 | 0.17 | 0.07 (0.04) | - - - | - - - |
| Monocyte CD14+ | 9.28 | 0.0001 | 0.0004 | 0.53 (0.52) | 2.48 (0.57) | - - - |
| T memory cell | 6.10 | 0.003 | 0.009 | -2.17 (0.92) | -3.13 (1.01) | - - - |
| log(Activated B) | 0.87 | 0.17 | 0.27 | 0.06 (0.05) | - - - | - - - |
| log(Monocyte CD14+CD16+) | 0.17 | 0.58 | 0.63 | 0.047 (0.08) | - - - | - - - |
| log(NK) | 2.87 | 0.08 | 0.16 | 0.11 (0.07) | 0.16 (0.08) | - - - |
| log(NKT) | 0.27 | 0.48 | 0.59 | -0.07 (0.10) | - - - | - - - |
| IL7α Receptor (n = 178) | 0.36 | 0.43 | 0.29 | 0.69 (0.87) | - - - | - - - |

Note: All estimates are from linear models with polynomials of exposure variable X, adjusted for age, BMI, ever smoked and effect of UAs/Cr

[a]$\Delta R^2$: The change in $R^2$ for percent of variation in outcome explained by the effect of PAH-DNA adducts adjusting for other variables.

[b]p-value was from likelihood ratio test for the effect of PAH-DNA adducts, possibly non-monotonic.

[c]FDR: False discovery rate

[d]$B_1$: Estimated coefficient of X; $B_2$: estimated coefficient of $X^2$, $B_3$: estimated coefficient of $X^3$; for X = log(PAH-DNA adducts /1.8357).

se: Standard error

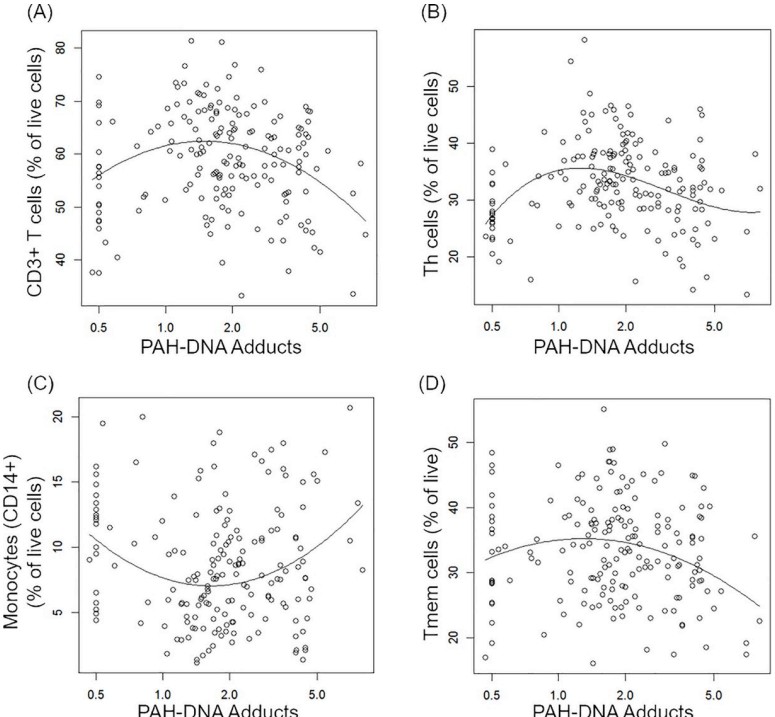

**Fig 2. Covariate-adjusted associations between PAHs and cell surface markers.** Individual data points represent observations; solid lines represent estimated mean outcome vs. PAH-DNA adducts adjusted for other variables in the model. The outcomes (A-D) were T (CD3+), Th (CD4+), monocytes (CD14+), and Tmem (CD3+CD45+) cells in logarithmic scale.

**Table 4. Association between urinary arsenic concentration per creatinine (UAs/Cr) and T cell subsets (n = 180).**

| T cell subsets | [a]$\Delta R^2$ (%) | [b]p-value | [c]FDR | [d]$B_1$ (se) | [d]$B_2$ (se) |
|---|---|---|---|---|---|
| Th cells | 2.22 | 0.04 | 0.10 | 1.47 (0.70) | --- |
| $(Th1\ cells)^{1/2}$ | 0.34 | 0.42 | 0.48 | -0.05 (0.06) | --- |
| $log(Th2\ cells)$ | <0.01 | 0.95 | 0.95 | 0.004 (0.07) | --- |
| Treg | 0.95 | 0.20 | 0.26 | -0.53 (0.41) | --- |
| $(Th17\ cells)^{1/3}$ | 6.10 | <0.001 | 0.004 | 0.13 (0.04) | --- |
| Stimulated CD3 | 1.21 | 0.13 | 0.22 | 1.40 (0.94) | --- |
| Stimulated CD4 | 4.40 | 0.02 | 0.08 | 1.09 (0.79) | 1.13 (0.61) |
| Stimulated Live | 1.56 | 0.095 | 0.19 | 1.82 (1.09) | --- |

All estimates are from linear models with polynomials of exposure variable X, adjusted for age, BMI, ever smoked and effect of PAH-DNA adducts

[a]$\Delta R^2$: The change in $R^2$ for percent of variation in outcome explained by the effect of UAscr adjusting for other variables.

[b]p-value was from likelihood ratio test for the effect of UAscr.

[c]FDR: False discovery rate

[d]$B_1$: Estimated coefficient of X; $B_2$: estimated coefficient of $X^2$; for X = log(UAscr /96.2963).

se: Standard error

For anti-CD3/anti-CD28 activated cells, we found that UAs/Cr was associated with a trend toward an increase in the percentage of CD4+ and activated CD4+ cells expressing CD25 and CD69. There was a moderate statistical association with As exposure and an increase in the percentage of Th17 cells. In contrast, PAH-DNA adducts associated with Th17 cells non-monotonically with a trend towards a decrease. At high levels of PAH-DNA adducts, we

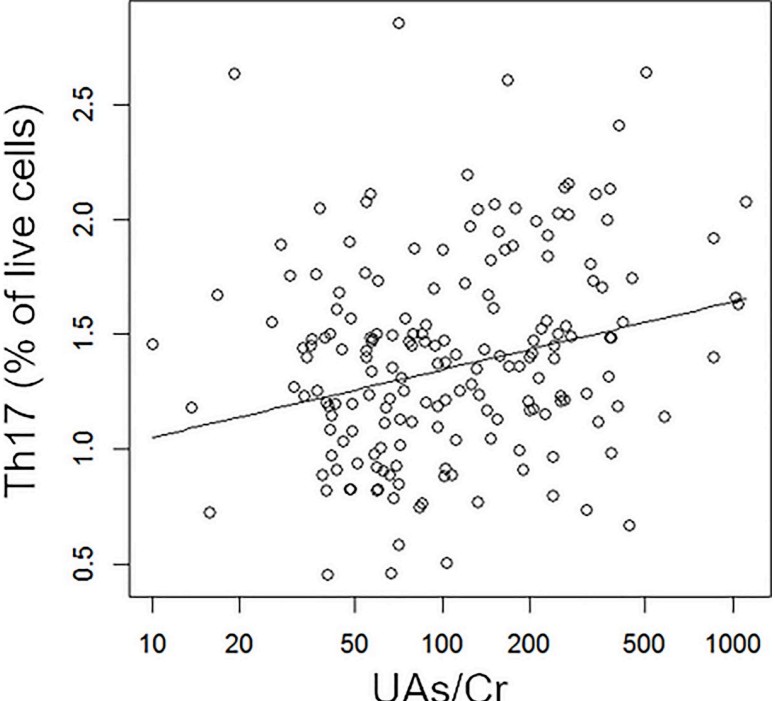

**Fig 3. Covariate-adjusted association between UAs/Cr and Th17 cells.** Individual data points represent observations; solid line represent estimated mean outcome $Th17^{1/3}$ vs. UAs/Cr adjusted for other variables in the model.

**Table 5. Association between PAH-DNA adducts and T cell subsets (n = 180).**

| T cell subsets | [a]$\Delta R^2$ (%) | [b]p-value | [c]FDR | [d]$B_1$ (se) | [d]$B_2$ (se) | [d]$B_3$ (se) |
|---|---|---|---|---|---|---|
| Th cells | 8.64 | 0.0003 | 0.0007 | -1.52 (0.98) | -4.47 (1.08) | --- |
| (Th1 cells)$^{1/2}$ | 10.92 | 0.0001 | 0.0006 | 0.53 (0.18) | 0.31 (0.10) | -0.25 (0.14) |
| log(Th2 cells) | 20.48 | <0.0001 | <0.0001 | 0.34 (0.20) | 0.55 (0.11) | -0.33 (0.15) |
| Treg cells | 0.87 | 0.22 | 0.35 | -0.68 (0.55) | --- | --- |
| (Th17 cells)$^{1/3}$ | 6.28 | 0.002 | 0.004 | -0.11 (0.05) | 0.11 (0.05) | --- |
| Stimulated CD3+ | 0.36 | 0.42 | 0.56 | -1.03 (1.27) | --- | --- |
| Stimulated CD4+ | 0.01 | 0.87 | 0.87 | 0.16 (1.01) | --- | --- |
| Stimulated Live | 0.04 | 0.78 | 0.87 | -0.41 (1.47) | --- | --- |

Note: All estimates are from linear models with polynomials of exposure variable X, adjusted for age, BMI, ever smoked and effect of UAs/Cr

[a]$\Delta R^2$: The change in $R^2$ for percent of variation in outcome explained by the effect of PAH-DNA adducts adjusting for other variables

[b]p-value was from likelihood ratio test for the effect of PAH-DNA adducts

[c]FDR: False discovery rate

[d]$B_1$: Estimated coefficient of X; $B_2$: estimated coefficient of $X^2$, $B_3$: estimated coefficient of $X^3$; for X = log(PAH-DNA adducts /1.8357).

se: Standard error

observed a significant decrease in the percentage of Th (CD4+) cells, and at lower levels of adducts non-monotonic associations appear to decrease and then increase in Th1 and Th2 cell populations. These findings demonstrate the complexity of the effects that As and PAH have on the immune system. The non-monotonic associations of As and PAH further complicate interpretation of findings in co-exposures.

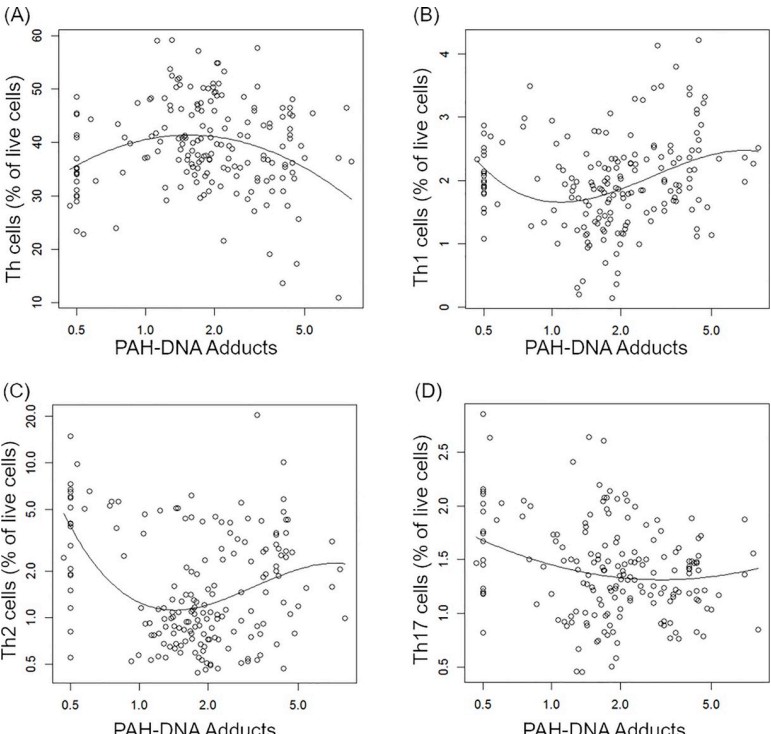

**Fig 4. Covariate-adjusted association between PAH-DNA adducts and Th, Th1, Th2 and Th17 cells.** Individual data points represent observations; solid lines represent estimated mean outcome vs. PAH-DNA adducts adjusted for other variables in the model. The outcome variables (A-D) were Th, Th1$^{1/2}$, log(Th2), and Th17$^{1/3}$.

Numerous studies have examined *in vitro* As exposure and immune function in animal models [21, 22, 27, 28, 40–43] and PBMC exposed *in vitro* [44–47]. In a review of the literature we did not find any studies that examined the association of UAs/Cr with CSM expressed by PBMC in adults. In children, Soto-Pena et al. [48] found a decrease in the ratio of Th-CD4 +/CTL CD8+ cells. Ahmed et al. [49] found a decrease in the immune response to purified protein derivative (PPD) that was associated with As exposure. Raqib et al. [50] found that UAs was associated with respiratory infection and diarrhea in children, and a decrease in humoral immunity and vaccine responses. An overall increase in childhood infections has also been reported [51].

Several human cord blood studies also examined the effects of As on cell subsets. Nadeau et al [52] found that As, measured as UAs, increased T cell proliferation in cord blood samples, a finding similar to our previous work [19]. In cord blood Nygaard et al. [53] found a decrease in Th and Tmem cells that was somewhat more prevalent in females than in males. Cord blood cells are highly enriched with hematopoietic stem cells and contain substantially more naïve T cells (CD45RA+) than are present in PBMC. Therefore, these studies may not be comparable to our work with adult males.

Based on animal and human *in vitro* PBMC studies, As is generally known as an immuno-suppressant, affecting several different signaling pathways in bone marrow [21, 27], thymus [22, 28], and HPBMC [44]. However, it is clear from our work that the effects of As are quite complex and may differ in PBMC cells obtained from adult males with chronic exposure. The increase in pro-inflammatory Th17 cells found in this study, and the increase in IL-1β produced by As and reported in our previous study [19] suggests that immune stimulation should also be considered when designing immunotoxicity studies in human cohorts exposed to As, and perhaps markers of autoimmune diseases should also be included.

Exposure to As is known to increase the risk of upper airway infection in children [54–56], suggesting that As may suppress local lung and airway immune cells not represented by changes in PBMC. We note that innate immune cells also play an important role in the lung [57]; however, these cells were not assessed in the present studies. Therefore, further work is needed to determine the use and limitations of PBMC CSM and activated cells for their prediction of infectious disease responses.

A limitation of this study was the discovery of similar PAH-DNA adduct profiles of smokers and non-smokers. The smokers had statistically significant amounts of PAH-DNA adducts; however, the overall the range of adducts was similar between the two groups. Another limitation was that only males were recruited for the study, due to the fact that there are inadequate numbers of female smokers in the HEALS cohort.

In summary, we found that PBMC obtained from men chronically exposed to As and/or PAHs demonstrated significant changes in immune cell subsets. These findings suggest that As may produce a pro-inflammatory environment in adult males. For PAH exposures, PAH-DNA adducts correlated with a decrease in several CSM, including T, Th, Tmem cells, and monocytes. Within the Th cells the subsets Th1, Th2, Th17, were associated with PAH exposure. PAH-DNA adducts and UAs had independent associations, suggesting that they may act via different mechanisms. This suggests that non-genotoxic signaling pathways, although sensitive to As, may not be predicted based on *in vitro* genotoxicity studies of DNA damage and As interactions with PAHs [58, 59].

## Supporting information

**S1 Fig. Gating strategy for cell surface markers.** Flow chart indicating the gating strategy for CSM. This flow chart has been modified from the original by Lauer et al. [34] to include

activated B cells.
(PDF)

**S2 Fig. Gating strategy for intracellular markers.** Flow chart indicating the gating strategy for ICS by Lauer et al. [35].
(PDF)

**S1 Table. Cell surface marker (CSM) antibodies for flow cytometry.**
(PDF)

**S2 Table. Intracellular marker (ICM) antibodies for flow cytometry.**
(PDF)

**S3 Table. Cell surface markers (CSM).**
(PDF)

**S4 Table. Intracellular staining (ICS).**
(PDF)

## Acknowledgments

We wish to acknowledge Professor Yu Chen and Dr. Fen Wu for review of this manuscript and other support. The data set for this study can be accessed at 10.6084/m9.figshare.8144492.

## Author Contributions

**Conceptualization:** Fredine T. Lauer, Faruque Parvez, Pam Factor-Litvak, Xinhua Liu, Regina M. Santella, Habibul Ahsan, Joseph Graziano, Scott W. Burchiel.

**Data curation:** Fredine T. Lauer, Faruque Parvez, Pam Factor-Litvak, Xinhua Liu, Regina M. Santella.

**Formal analysis:** Fredine T. Lauer, Faruque Parvez, Pam Factor-Litvak, Xinhua Liu, Regina M. Santella, Tariqul Islam, Mahbubul Eunus, Scott W. Burchiel.

**Funding acquisition:** Fredine T. Lauer, Faruque Parvez, Pam Factor-Litvak, Xinhua Liu, Regina M. Santella, Habibul Ahsan, Joseph Graziano, Scott W. Burchiel.

**Investigation:** Fredine T. Lauer, Faruque Parvez, Pam Factor-Litvak, Xinhua Liu, Regina M. Santella, Tariqul Islam, Mahbubul Eunus, Nur Alam, A. K. M. Rabiul Hasan, Mizanour Rahman, Habibul Ahsan, Joseph Graziano, Scott W. Burchiel.

**Methodology:** Fredine T. Lauer, Pam Factor-Litvak, Xinhua Liu, Regina M. Santella, Tariqul Islam, Joseph Graziano, Scott W. Burchiel.

**Project administration:** Fredine T. Lauer, Faruque Parvez, Xinhua Liu, Regina M. Santella, Tariqul Islam, Mahbubul Eunus, Nur Alam, A. K. M. Rabiul Hasan, Mizanour Rahman, Joseph Graziano, Scott W. Burchiel.

**Resources:** Faruque Parvez, Scott W. Burchiel.

**Supervision:** Fredine T. Lauer, Faruque Parvez, Pam Factor-Litvak, Xinhua Liu, Nur Alam, Joseph Graziano, Scott W. Burchiel.

**Validation:** Fredine T. Lauer, Faruque Parvez, Pam Factor-Litvak, Xinhua Liu, Regina M. Santella, Joseph Graziano, Scott W. Burchiel.

**Writing – original draft:** Fredine T. Lauer, Faruque Parvez, Pam Factor-Litvak, Xinhua Liu, Regina M. Santella, Habibul Ahsan, Joseph Graziano, Scott W. Burchiel.

**Writing – review & editing:** Fredine T. Lauer, Faruque Parvez, Pam Factor-Litvak, Xinhua Liu, Regina M. Santella, Mahbubul Eunus, Nur Alam, Habibul Ahsan, Joseph Graziano, Scott W. Burchiel.

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
