## [Decision Letter · Decision Letter 0]

9 Jul 2019

PONE-D-19-15877

Changes in human peripheral blood mononuclear cell (HPBMC) populations and T-cell subsets associated with arsenic and polycyclic aromatic hydrocarbon exposures in a Bangladesh cohort

PLOS ONE

Dear Dr. Burchiel,

Thank you for submitting your manuscript to PLOS ONE. After careful consideration, we feel that it has merit, but does not fully meet PLOS ONE’s publication criteria as it currently stands. Reviewers were generally enthusiastic about the study, and only have a few minor comments/suggestions. Therefore, we invite you to submit a revised version of the manuscript that addresses all the points raised by the reviewers.

We would appreciate receiving your revised manuscript by August 25, 2019. To enhance the reproducibility of your results, we recommend that if applicable you deposit your laboratory protocols in protocols.io, where a protocol can be assigned its own identifier (DOI) such that it can be cited independently in the future. For instructions see: http://journals.plos.org/plosone/s/submission-guidelines#loc-laboratory-protocols

We look forward to receiving your revised manuscript.

Kind regards,

M. Firoze Khan, Ph.D.

Academic Editor

PLOS ONE

Journal Requirements:

Reviewers' comments:

Reviewer's Responses to Questions

**Comments to the Author**

1. Is the manuscript technically sound, and do the data support the conclusions?

Reviewer #1: Yes

Reviewer #2: Yes

Reviewer #3: Yes

2. Has the statistical analysis been performed appropriately and rigorously? 

Reviewer #1: Yes

Reviewer #2: I Don't Know

Reviewer #3: Yes

3. Have the authors made all data underlying the findings in their manuscript fully available?

Reviewer #1: Yes

Reviewer #2: Yes

Reviewer #3: Yes

4. Is the manuscript presented in an intelligible fashion and written in standard English?

Reviewer #1: Yes

Reviewer #2: Yes

Reviewer #3: Yes

5. Review Comments to the Author

Reviewer #1: Both arsenic (As) and polycyclic aromatic hydrocarbons (PAH) are common environmental contaminants. The manuscript titled “Changes in human peripheral blood mononuclear cell (HPBMC) populations and T-cell subsets associated with arsenic and polycyclic aromatic hydrocarbon exposures in a Bangladesh cohort” examined associations between As and PAH exposures with lymphoid cell populations in human peripheral blood mononuclear cells (PBMC), as well as alterations in differentiation and activation of B and T cells. Even though the authors did not find evidence of interactions between As and PAH exposures, they obtained a large amount of interesting findings such as the associations between individual exposures (As or PAH) and changes to immune cell populations (particularly Th cells) and immune cell activation. The findings are useful in understanding the mechanisms of As- or PAH-mediated adverse health outcomes, such as immune suppression or autoimmunity

This manuscript should be suitable for publication in Plos One if the minor issues below are addressed:

1) The authors may consider reducing the length of method section. Some methods such as collection of PBMC, cell surface and intracellular staining are common techniques and should be described briefly with appropriated references.

2) The authors may need to provide information on why they selected the current exposure biomarkers (UAs and PAH-DNA adducts) in this study.

Reviewer #2: This is a very written manuscript and detailed, although methodology can be shortened with relevant citations rather than extensive description. Arsenic exposure in drinking water in Bangladesh is a genuine heath problem along with PAH exposure through cigarette smoking. As described on page 5, goal of the study was to examine the combined effect of As and PAH, where half of the participants were identified drinking from the wells with water contamination of As >50 ppb and remainder with <50 ppb with or without smoking. I for see four study groups or more in the "Recruitment and consent of study participants" section, which needs to be translated in the tables and figures so that comparisons can be understood. Otherwise, this is a significant study addressing an important health issue in a population group ignorant of serious adverse health effects of As and PAH.

Reviewer #3: In this investigation, the authors have investigated the associations between arsenic (As) and PAH exposures with lymphoid cell populations in human peripheral blood mononuclear cells (PBMC), as well as alterations in differentiation and activation of B and T cells.. The authors have analyzed blood and urine from 200 men participating in the Health Effects of Arsenic Longitudinal Study (HEALS) in Bangladesh were selected for the present study based on their exposure to As from drinking water and their cigarette smoking status. The major findings were; (i) There was no evidence of interactions between As and PAH exposures; however, individual exposures (As or PAH) were associated with changes to immune cell populations, including Th cell subsets; (ii) As exposure was associated with an increase in the percentage of Th cells, and dose dependent changes in monocytes, NKT cells and a monocyte subset, and within the Th cell subset the authors found that arsenic exposure was also associated with a significant increase in the percentage of circulating pro-inflammatory Th17 cells; and (iii) PAH exposure was associated with changes in T cells, monocytes and T memory (Tmem) cells and with changes in Th, Th1, Th2 and Th17 subsets all of which were dose dependent. Based on these observations, the authors suggest that alterations of immune cell populations caused by environmental exposures to As and PAH may result in adverse health outcomes, such as changes in systemic inflammation, immune suppression, or autoimmunity. The manuscript is well written, and study appears to have been well conducted. This work is highly significant because little data exists regarding the associations between combined exposures to As and PAH with immunotoxicity in humans I have a few minor concerns that need to be addressed:

1. In the abstract (line 29 ) the sentence should not start with a number.

2. Since cigarette smoke contains over 4000 chemicals in addition to PAHs and As, the authors should include in the discussion the possible contribution of other chemicals to the changes in systemic inflammation and autoimmunity.

6. PLOS authors have the option to publish the peer review history of their article (what does this mean?). If published, this will include your full peer review and any attached files.

Reviewer #1: No

Reviewer #2: No

Reviewer #3: Yes: Bhagavatula Moorthy

---

## [Author Response · Author response to Decision Letter 0]

11 Jul 2019

Review Comments to the Author

Reviewer #1: Both arsenic (As) and polycyclic aromatic hydrocarbons (PAH) are common environmental contaminants. The manuscript titled “Changes in human peripheral blood mononuclear cell (HPBMC) populations and T-cell subsets associated with arsenic and polycyclic aromatic hydrocarbon exposures in a Bangladesh cohort” examined associations between As and PAH exposures with lymphoid cell populations in human peripheral blood mononuclear cells (PBMC), as well as alterations in differentiation and activation of B and T cells. Even though the authors did not find evidence of interactions between As and PAH exposures, they obtained a large amount of interesting findings such as the associations between individual exposures (As or PAH) and changes to immune cell populations (particularly Th cells) and immune cell activation. The findings are useful in understanding the mechanisms of As- or PAH-mediated adverse health outcomes, such as immune suppression or autoimmunity

This manuscript should be suitable for publication in Plos One if the minor issues below are addressed:

1. The authors may consider reducing the length of method section. Some methods such as collection of PBMC, cell surface and intracellular staining are common techniques and should be described briefly with appropriated references. 

Response: we have significantly reduced the length of the Methods section.

2. The authors may need to provide information on why they selected the current exposure biomarkers (UAs and PAH-DNA adducts) in this study.

Response: We have measured other exposure markers in the study as well and decided to focus on UAs and PAH-adduct for this manuscript because we found them to be most reliable for associations with immune endpoints. The UAs is highly correlated with the home water As, another indicator of long term exposure used in epi studies. We chose UAs as we know that the men in the study worked away from their homes and consumed significant amounts of water from sources other than their home. The UAs is also obtained at the same time as the blood sample draws for study, so we feel it is a better measure of exposure in this setting. We chose PAH-DNA adducts (BPDE-like) as they are a good measure of exposure to large PAHs, like BaP. We also have significant experience with these large PAHs and we know that they produce immune suppression.

Reviewer #2: This is a very written manuscript and detailed, although methodology can be shortened with relevant citations rather than extensive description. Arsenic exposure in drinking water in Bangladesh is a genuine heath problem along with PAH exposure through cigarette smoking. As described on page 5, goal of the study was to examine the combined effect of As and PAH, where half of the participants were identified drinking from the wells with water contamination of As >50 ppb and remainder with <50 ppb with or without smoking. I for see four study groups or more in the "Recruitment and consent of study participants" section, which needs to be translated in the tables and figures so that comparisons can be understood. - Otherwise, this is a significant study addressing an important health issue in a population group ignorant of serious adverse health effects of As and PAH.

Response:

1. We have shortened the methods

2. As pointed out by the reviewer, there were four study groups used for recruitment in a 2 x 2 design when the study was implemented. This design assumed that cigarette smokers would have substantially higher PAH-DNA adducts than never smokers. In our analysis, other sources of environmental PAH exposures were so high that there was only a small (but significant) increase in cigarette smoker PAH-DNA adducts. We found that many of the effects of cigarette smoking were independent of PAH-DNA adducts, presumably due to the presence of many other chemicals in cig smoke (see Reviewer 3 comment) that were not associated with BaP-like exposures and measured in our PAH-DNA adduct assay. The Reviewer suggested a 2 x 2 analysis and presentation. However, due to the complexity of these effects, we are examining cigarette smoke-As interactions in great detail and we will report these findings separately.

Reviewer #3: In this investigation, the authors have investigated the associations between arsenic (As) and PAH exposures with lymphoid cell populations in human peripheral blood mononuclear cells (PBMC), as well as alterations in differentiation and activation of B and T cells.. The authors have analyzed blood and urine from 200 men participating in the Health Effects of Arsenic Longitudinal Study (HEALS) in Bangladesh were selected for the present study based on their exposure to As from drinking water and their cigarette smoking status. The major findings were; (i) There was no evidence of interactions between As and PAH exposures; however, individual exposures (As or PAH) were associated with changes to immune cell populations, including Th cell subsets; (ii) As exposure was associated with an increase in the percentage of Th cells, and dose dependent changes in monocytes, NKT cells and a monocyte subset, and within the Th cell subset the authors found that arsenic exposure was also associated with a significant increase in the percentage of circulating pro-inflammatory Th17 cells; and (iii) PAH exposure was associated with changes in T cells, monocytes and T memory (Tmem) cells and with changes in Th, Th1, Th2 and Th17 subsets all of which were dose dependent. Based on these observations, the authors suggest that alterations of immune cell populations caused by environmental exposures to As and PAH may result in adverse health outcomes, such as changes in systemic inflammation, immune suppression, or autoimmunity. The manuscript is well written, and study appears to have been well conducted. This work is highly significant because little data exists regarding the associations between combined exposures to As and PAH with immunotoxicity in humans I have a few minor concerns that need to be addressed:

1. In the abstract (line 29 ) the sentence should not start with a number. 

Response: we have made the change.

2. Since cigarette smoke contains over 4000 chemicals in addition to PAHs and As, the authors should include in the discussion the possible contribution of other chemicals to the changes in systemic inflammation and autoimmunity. 

Response: We agree with the reviewer and have addressed this in some detail in the response to Reviewer 2 above. We have also added a sentence in the Discussion (marked line 481, final line 449-452 ) to address the potential role of arsenic in immune stimulation. The effects of other chemicals in cig smoke were not assessed in the present study, but are considered in our future analysis.

---

## [Editor Report · Decision Letter 1]

17 Jul 2019

Changes in human peripheral blood mononuclear cell (HPBMC) populations and T-cell subsets associated with arsenic and polycyclic aromatic hydrocarbon exposures in a Bangladesh cohort

PONE-D-19-15877R1

Dear Dr. Burchiel,

We are pleased to inform you that your manuscript has been judged scientifically suitable for publication and will be formally accepted for publication once it complies with all outstanding technical requirements.

With kind regards,

M. Firoze Khan, Ph.D.

Academic Editor

PLOS ONE
---

## [Editor Report · Acceptance letter]

19 Jul 2019

PONE-D-19-15877R1 

Changes in human peripheral blood mononuclear cell (HPBMC) populations and T-cell subsets associated with arsenic and polycyclic aromatic hydrocarbon exposures in a Bangladesh cohort 

Dear Dr. Burchiel:

I am pleased to inform you that your manuscript has been deemed suitable for publication in PLOS ONE. Congratulations! Your manuscript is now with our production department. 

With kind regards,

on behalf of

Dr. M. Firoze Khan 

Academic Editor

PLOS ONE